# The Influence of Dietary Supplementations on Neuropathic Pain

**DOI:** 10.3390/life12081125

**Published:** 2022-07-27

**Authors:** Francesco D’Egidio, Giorgia Lombardozzi, Housem E. Kacem Ben Haj M’Barek, Giada Mastroiacovo, Margherita Alfonsetti, Annamaria Cimini

**Affiliations:** 1Department of Life, Health and Environmental Sciences, University of L’Aquila, 67100 L’Aquila, Italy; francesco.degidio@graduate.univaq.it (F.D.); giorgia.lombardozzi@student.univaq.it (G.L.); houseneddine.kacembenhajmbarek@student.univaq.it (H.E.K.B.H.M.); giada.mastroiacovo@student.univaq.it (G.M.); margherita.alfonsetti@guest.univaq.it (M.A.); 2Sbarro Institute for Cancer Research and Molecular Medicine and Center for Biotechnology, Temple University, Philadelphia, PA 19122, USA

**Keywords:** CIPN, diabetic neuropathy, neuropathic pain, nutraceuticals, nutritional supplement, gut–brain axis

## Abstract

Neuropathic pain is defined as pain caused by a lesion or disease of the somatosensory nervous system and affects 7–10% of the worldwide population. Neuropathic pain can be induced by the use of drugs, including taxanes, thus triggering chemotherapy-induced neuropathic pain or as consequence of metabolic disorders such as diabetes. Neuropathic pain is most often a chronic condition, and can be associated with anxiety and depression; thus, it negatively impacts quality of life. Several pharmacologic approaches exist; however, they can lead numerous adverse effects. From this perspective, the use of nutraceuticals and diet supplements can be helpful in relieve neuropathic pain and related symptoms. In this review, we discuss how diet can radically affect peripheral neuropathy, and we focus on the potential approaches to ameliorate this condition, such as the use of numerous nutritional supplements or probiotics.

## 1. Introduction

Neuropathic pain is a chronic condition caused by lesion or pathologies of the somatosensory nervous system with damage and loss of function of nerve cells. It can be caused by a variety of stimuli, in particular metabolic diseases and chemotherapeutic treatments. Neuropathic pain is a state characterized by a broad spectrum of symptoms, including paresthesia and dysesthesia manifested as numbness, tingling, and altered touch perception, as well as mechanical or thermal allodynia and hyperalgesia, which are able to negatively affect the quality of life. It affects from 7% to 10% of the world population. To date, several therapeutic strategies to alleviate the symptoms have been studied. From a pharmacological point of view, despite their side effects, the anticonvulsants gabapentin and pregabalin, tricyclic antidepressants and serotonin–noradrenaline reuptake inhibitors are used as first-line treatment. However, the pharmacological approach is unable to fully counteract the neuropathic pain condition. Therefore, several new and different approaches for the treatment of neuropathic pain have been proposed, including nerve stimulation techniques—such as transcutaneous electrical nerve stimulation and spinal cord or deep brain stimulation, acupuncture, surgical nerve decompression—as well as behavioral, cognitive and animal therapies, exercise and diaphragmatic breathing, nutritional supplements and diet therapy. Among them, the therapy based on diet and nutritional supplements or nutraceuticals was shown to be able to relieve pain, especially when this condition is poorly responsive to drugs or therapeutic options are limited or characterized by strong side effects. In recent years, nutraceutical and other non-pharmacological supplements have become highly relevant due to their synergistic activity with other ongoing pharmacological and non-pharmacological therapies in counteracting neuropathic pain. For example, ω-3 fatty acids concomitantly administered with paclitaxel showed a protective activity in chemotherapy-induced neuropathic pain (CIPN) patients [1]. Additionally, B vitamins administration in patients with pain syndromes of the lumbar spine treated with diclofenac led to reduction in pharmacological therapy dosage due to strong pain relief [2]. However, the lack of clinical studies is remarkable. So far, preclinical research on this topic collected a large amount of data, but many aspects of nutraceuticals remain unclear.

Even if their mechanism of action is not well-understood, nutraceuticals and non-pharmacological supplements have emerged as treatments for neuropathic pain in various conditions, including diabetic neuropathy (DN) and CIPN. In this review, the role of nutritional supplements in the treatment of neuropathic pain caused by metabolic diseases or chemotherapeutics is explored.

## 2. Neuropathic Pain

Neuropathic pain is a chronic state characterized by a wide spectrum of symptoms and signs occurring after a primary lesion or pathology of the somatosensory nervous system. Among them are allodynia, hyperalgesia, and paresthesia with tingling, itching, and loss of sensitivity. In patients suffering from neuropathic pain, the perceived pain is usually spontaneous, manifesting itself without a stimulus and negatively affecting the patient’s quality of life [3]. The pathology that causes neuropathic pain particularly involves the small unmyelinated C fibers and the myelinated A fibers, i.e., the Aβ and Aδ fibers [1]. The following pain can be focal or diffuse, with peripheral nerve damage (peripheral neuropathic pain) or pathologies or lesions of the central nervous system (central neuropathic pain), always characterized by an abnormal sensation of pain and sensory deficit in the same area of the skin over time. The localization of the pain in these conditions typically regards the distal extremities in a distribution often known as “glove and stocking” due to the major involvement of feet, calves, hands and forearms [4,5]. Neuropathic pain shows the coexistence of positive and negative somatosensory signs. While negative signs are characterized by sensory deficits to noxious and thermal stimuli, indicating lesions with small-diameter afferent fibers (C-, Aδ-fibers), positive signs include both non-painful and painful sensations, such us, for example, non-painful paresthesia and spontaneous or stimulus-induced pain [6]. Among the numerous conditions and pathophysiological states that determine the onset of neuropathic pain, the most relevant are DN and CIPN, both a consequence of a peripheral insult [4]. DN is a frequent complication of long-term diabetes. It is characterized by comorbidities, such as sleep disorders, anxiety and depression, severe pain and cardiovascular diseases, and increasing mortality rate in patients [7]. Similarly, CIPN is a common pain condition following chemotherapeutic treatments. Chemotherapeutics administration often results in neurotoxicity through drug-related mechanisms, leading to acute or chronic symmetrical sensory neuropathy and autonomic dysfunction. CIPN is usually mild and reversible, but in other cases, this neuropathy can be irreversible and severe, strongly interfering with daily activities and the chemotherapeutic regimen [8]. To date, the treatment for those conditions spans pharmacological and non-pharmacological therapies. However, the current therapies are not so effective and are usually followed by several side effects. Therefore, the absence of specific treatments for neuropathic conditions such as DN and CIPN emphasized the exploration of new strategies to restore patient’s quality of life. Combinations of pharmacological and non-pharmacological therapies have been proposed as a possible strategy to counteract pain with less side effects. In this context, nutraceuticals and non-pharmacological supplements, as well as a specific diet, are proposed as valid adjuvant therapies in neuropathic pain conditions, such as DN and CIPN [9].

### 2.1. Epidemiology

Neuropathic pain is a common condition that dramatically affects the quality of life in patients. This condition has a prevalence of 7–10% in the world population. Although the recording of incidence and prevalence is difficult in the general population due to the current classification system, it is clear how theyare going to increase [10,11,12]. This is due to the increasing rate of obesity and survival of cancer patients treated with chemotherapeutics [13]. To date, classification is made following the International Classification of Disease, which is not focused on the pain condition but only on the damage or the disease underlying the pain, as consequence of the high variability of the association of underlying neurological disease and pain [12]. In fact, pain is not always the major manifestation following neurological disease. In CIPN or DN pain may occur only in a subset of patients, depending on the studied population and the diagnostic tools [7,14]. The administration of chemotherapeutic agents is often followed by neuropathic pain. It was found that 68% of patients shows CIPN during the first month after cancer therapy and 60% within 3 months, depending on the chemotherapeutic agent, dosing regimen, and clinical conditions. The 30% of patients showed CIPN at 6 months and beyond, acquiring the chronic stage of CIPN [15]. However, it is difficult to predict outcomes due to the absence of validated biomarkers and the high heterogeneity of CIPN risk [16]. Given a worldwide prevalence of 7–10% for neuropathic pain, DN is the most common form of neuropathy worldwide. Its prevalence increases with the duration of diabetes and the increment of diabetes prevalence, spanning from 6% to 51% in function of the studied population [7,17]. For example, of all the patients with diabetes mellitus, about 20–50% of the patients suffer from DN, with neuropathic pain as one of the major symptoms. However, 25% of diabetes patients are characterized by the absence of neuropathy even in the presence of painful symptoms [10,18,19,20,21].

### 2.2. Pathophysiology

Neuropathic pain is caused by lesions or pathologies of the somatosensory nervous system, leading to the deregulated transmission of sensory signals into the spinal cord after peripheral injury. This is due to the variation, within the damaged nerve, of the activity of ion channels. In the absence of neuropathic pain, normal pain circuitries cause the activation of a nociceptor in response to a painful stimulus, causing depolarization to reach first-order neurons with entrance of sodium via sodium channels and the release of potassium [22]. In the ending of first-order neurons at spinal cord level, voltage-gated calcium channels are opened by electrochemical signals in the pre-synaptic terminal, allowing the entrance of calcium. This leads to glutamate release into the synaptic space allowing it to bind to N-methyl-D-aspartate (NMDA) receptors on second-order neurons. The synaptic transmission is determined by the reach of an excitatory post-synaptic potential. In fact, glutamate interacting with NMDA receptors causes ion influx increasing post-synaptic neuron excitability [23]. From the spinal cord, second-order neurons reach the thalamus, where synapses with third-order neurons connected to the limbic system and cerebral cortex. A pain inhibitory pathway also exists in the dorsal horn. Anti-nociceptive neurons originating in the brain stem travel down the spinal cord. Then, they synapse with short interneurons in the dorsal horn. The releasing of serotonin and norepinephrine allow interneurons to modulate the synapse between the first-order neurons and the second-order neurons. This is possible due to gamma-aminobutyric acidergic (GABAergic) and enkephalinergic fibers originating in the brain stem rostral ventral medulla that are responsible of gamma aminobutyric acid and enkephalin release, leading to the presynaptic inhibition of primary afferent neurons [24,25]. Thus, pain prevention is the result of the inhibition of synapses activity between first- and second-order neurons [26]. In neuropathic pain, there is an increased excitability caused by the intensified expression and activity of sodium channels, with the loss of potassium channels that are usually activity modulators with an increment of transmitters release at the spinal central ending of the sensory nerves [22,27]. The neurodegeneration caused by nerve injury leads to the disruption of the connection between the periphery and central nervous systems, ending in complete sensory loss. In fact, distal axons die because of Wallerian degeneration after the disruption of axons from primary sensory neurons [28]. Moreover, at the injury site, the proximal residues of fiber, such as C-fibers, can generate ectopic activity causing pain to start from an area with reduced sensitivity to mechanical and thermal stimuli. This spontaneous activity generates ongoing pain due to ectopic action potential through an enhanced synaptic transmission to the spinal neurons or enhanced intrinsic excitability of second-order neurons. Ongoing pain can also occur at multiple sites along the axon and in the dorsal root ganglia of nociceptors [29,30]. Hence, an increased magnitude of response to spontaneous activity and noxious stimulation are consequences of sensitized nociceptors such as unmyelinated (C) and thinly myelinated (Aδ) primary afferent neurons [28].

From a molecular point of view, nociceptive physical and chemical stimuli are able to activate transient receptor potential (TRP) channels, resulting in production of the potential need to activate voltage-gated sodium channels, resulting in ectopic discharge. In particular, a variety of conditions (for example, inflammation and inflammatory microenvironment, acidic pH and body temperature) lead to the enhancement of Transient Receptor Potential Vanilloid 1 (TRPV1) channel function via phosphorylation, causing the translocation of TRPV1 to the cell surface with the production of the action potential and discharge responsible for peripheral sensitization [4,31,32]. In addition to this, the sensitization of remaining intact axons can be obtained through the infiltration of immune cells, such as macrophages, neutrophils, Th1 and Th6 cells, and mast cells, at the injured site. In this environment, the accumulation of immune cells and the release of substances—such as neuropeptides from nociceptive axons, nitric oxide (NO), the nerve growth factor (NGF) and matrix metalloproteinases (MMPs) from Schwann cells, kinins, ATP, lipids prostaglandins, cytokines—contribute to axonal damage and cause a local increase in blood flow and tissue swelling [33]. In particular, a relevant role has been suggested for arachidonic acid in neuropathic pain inflammation, leading to cascades of pro-inflammatory cytokines, such as interleukins 1β (IL-1β), 6 (IL-6), and 8 (IL-8), Interferon **γ** (INF**γ**), Tumor Necrosis Factor α (TNFα), [34,35,36]. Hence, the mechanism behind the damage of nervous system structures involves mainly oxidative stress and altered ion channel activity, but also mitochondrial damage, microtubule disruption, myelin sheath damage, DNA damage, neuroinflammation and other immunological processes [7,37,38].

## 3. Nutritional Supplements in Neuropathic Pain

The increasing concern about pain management in neuropathic pain conditions caused by the absence of specific pharmacological treatment led to an exploration of the possibility of combined therapy between pharmacological and non-pharmacological compounds. When pain is not alleviated by prescribed pharmacologic agents or when those drugs are not easily accessible, researchers and patients look for alternative therapies. In this context, even if their mechanism of action is not completely understood, nutritional supplements were suggested to treat chronic illnesses that are poorly responsive to drugs, such as neuropathic pain, including DN and CIPN [9,39]. In recent years, several studies have shown how higher intakes of vegetables and fruit that are rich in antioxidants improve the function of the immune system and fight free radical damage, one of the major molecular actors in neuropathic pain. A supplement can be defined as a product used to support the diet and that contains one or more of the following compounds: vitamins, minerals, herbs or other botanicals, amino acids, live microbials, dietary substances used to increase the total dietary intake or a concentrate, metabolites, constituents, extracts, or combination of any of these ingredients [40]. To study neuropathic pain and observe the effects of nutritional supplements on pain conditions, several animal models, in particular mice, were developed. Neuropathic pain is induced in mice through peripheral nerve injury carried out via axotomy or chronic constriction injury, through drug-induced injury models administering anti-cancer agents, such as paclitaxel, or through disease-induced models with streptozotocin to induce DN [41,42]. Even so, knowledge about clinical efficacy and mechanism of action is still limited. It is necessary to establish the role of these compounds in neuropathic pain via large randomized controlled trials. Here, we explored the nutritional supplements for the treatment of neuropathic pain, including vitamins (B, C, D), minerals (zinc, magnesium), botanicals (*Boswellia serrata*, *Hypericum perforatum*, curcumin, capsaicin, menthol, bromelain), amino acids (acetyl-L-carnitine, acetyl-L-cysteine), fatty acids (ω-3 fatty acids, *N*-palmitoylethanolamide, α-lipoic acid), and probiotics (Table 1).

### 3.1. Vitamins in Neuropathic Pain

Among all the vitamins of B family, thiamine (B1), pyridoxine (B6), folate (B9), and cyanocobalamin (B12) are known to play extremely relevant role in a variety of processes, such as DNA and RNA synthesis, metabolism, immunity and neuroprotection [100,101]. Numerous studies have suggested the protective role of B vitamins in alleviating neuropathic pain, in particular in DN and CIPN. This has been tested in animal and human models of different neuropathic pain conditions. For example, the administration of a cocktail of vitamins B1, B6, and B12 to diabetic rats was found to improve tactile allodynia in diabetic. In the same model, sensory nerve conduction was improved via the administration of vitamin B6 [102]. Moreover, it was suggested that a stronger reduction in pain caused by DN can be obtained through higher dosages of thiamine and pyridoxine (25 and 50 mg/day) compared with lower doses (1 mg/day of each vitamin) [43]. This confirms the potential use of combination of B vitamins in the treatment of DN [44,45]. Regarding folate, its role as a methylator in the nervous system led to the exploration of its role in neuropathic pain. In a mouse model of neuropathic pain with spinal cord injury the treatment with folate significantly reduced thermal hyperalgesia compared with control mice. Additionally, a reduction In MMP2 has been found after treatment with folate. MMP2s are involved in the induction of neuropathic pain, so a potential mechanism for the counteracting of pain using folate was suggested [46]. On the contrary, the decreased level of B vitamins has been hypothesized as risk factor for CIPN. In a double-blinded randomized controlled trial vitamin B6 has been able to reduce CIPN from cisplatin and hexamethylenamin administration. However, further results indicate an alteration in response duration after the administration of high dose of vitamin B6 [103]. The mechanism of action of vitamin B in treating neuropathic pain is still unclear. A potential role of vitamin B12 has been proposed. It should be implicated in nerve repair and myelination, and thus may improve neuropathy symptoms. However, randomized trials of vitamin B for neuropathic pain are needed to assess the mechanisms and effects of specific type of vitamin B [47,48].

Vitamin C, also called ascorbic acid, plays different relevant roles, from collagen synthesis to improving non-heme iron’s absorption, with immunological function and neuroprotection. Vitamin C is also involved in the maintenance of the integrity of vascular cells. It is synthetized from glucose through the glucuronic acid pathway in animals. Humans lost this biosynthetic ability during evolution. Thus, the intakes of vitamin C in diet are necessary. Vitamin C deficiency causes bleeding knees, wrists and ankles, with arthralgia and myalgia, as manifestations of scurvy disease [104,105]. The use of vitamin C is shown to be effective in several aspects of pain management. In the molecular context of neuropathic pain, vitamin C may play a role in the reduction of reactive oxygen species stimulating the activity of NADPH oxidase and by acting as scavenger of oxidant species [101,106]. Several studies confirm the efficacy of vitamin C in reducing pain derived from neuropathic pain conditions. In particular, a total of 300 patients with type II diabetic and newly diagnosed DN, of either gender, were enrolled in an open-label, parallel-arm, interventional study. Patients were followed up for up to 12 weeks. At the end of the study the reduction in pain in the vitamin-treated group was remarkable [54]. Again, in a randomized control trial, the supplementation of vitamin C in a smaller group of DN’s patients (75 patients) reduced the pain condition that characterizes DN [55].

Vitamin D is a hormone that is synthesized in the skin by sunlight and plays a relevant role in calcium homeostasis. It may be involved in a large number of mechanisms underlying pain development. This is suggested by the diffuse localization in central nervous system of vitamin D receptor and 1α-hydroxylase enzyme, which convert the vitamin D’s pro-form into its active form [107,108]. Although there is no clear mechanism of action of vitamin D, one proposed explanation is that vitamin D is involved in the regulation of many processes implicated in pain manifestation: upregulation of Transforming Growth Factor α1, IL-4 and TNF-α; influence on prostaglandin action interacting with Cyclooxygenase 2, 15-Hydroxyprostaglandin dehydrogenase and Prostaglandin E2; neuroprotection via the increment of synthesis of neurotrophins and inhibition of inducible nitric oxide synthase [57,109,110]. Treatment with vitamin D has been explored in chronic neuropathic pain conditions, including DN and CIPN. Moreover, these conditions are characterized by vitamin D deficiency [57]. Vitamin D supplementation has been studied in patients with diabetes mellitus [58,59]. In two different prospective studies, the ability of vitamin D to counteract pain derived from DN caused by diabetes mellitus was observed [60,61]. In a smaller, randomized, placebo-controlled trial of 57 patients with DN, a significant decrease in pain scores in the treatment group compared with the placebo was observed [62]. Regarding CIPN, the depletion of vitamin D was proposed as risk factor for such condition. The usage of vitamin D as a pre-chemotherapy supplementation with neuroprotective effect in CIPN prophylaxis has been also suggested [63]. However, the real efficacy of vitamin D for neuropathic pain treatment remains unclear. Randomized controlled trials are needed to test this treatment in order to assess role and mechanisms of vitamin D in the treatment of neuropathic pain conditions.

### 3.2. Minerals in Neuropathic Pain

Minerals are natural elements that play an essential role in diet. They can be found in food and drinking water and are vital for all living organisms to maintain physical health. This is due to their involvement in many processes, such as hormone synthesis, bone formation, antioxidant activities, and others. Among all the mineral, two of them have been suggested as potential therapeutic instrument against neuropathic pain conditions, among which DN and CIPN: zinc and magnesium.

Zinc is an essential micronutrient used as treatment for different diseases, such as alcohol-related liver disease, sickle cell anemia, macular degeneration, and neuropathic pain [111,112]. Zinc plays a relevant role as an antioxidant compound in the activation of hundreds of genes involved in proliferation and cell growth. The deficiency of this mineral is associated with DNA damage, immune suppression, and apoptosis, ultimately leading to acrodermatitis enteropathica, a fatal condition if untreated [112,113,114]. A pain state is usually associated with inflammation, and zinc is a metallothionein with anti-inflammatory properties. Those properties are suggested to be responsible for the therapeutic benefit of zinc [115]. The efficacy of zinc in reducing pain has been assessed in a variety of animal models. In rats with induced neuropathic pain, the supplementation of zinc induced a reduction in pain. From a molecular point of view, zinc caused a decrease in NGF, involved in sensory hypersensitivity, and IL-1β levels, a marker of inflammation [116,117]. In another study, streptozotocin-induced diabetic rats were treated with zinc in order to establish the role and efficacy of minerals in counteracting DN symptoms. A slight restoration of motor nerve conduction speed, a reduction in tactile response threshold in diabetic rats and the counteraction of oxidative stress were was observed, suggesting the protective role of zinc supplementation in rat models of DN [118]. Similarly, in a mice model of CIPN treated with paclitaxel, zinc showed a protective role, reducing local allodynia. This is due to the inhibition of the TRPV1 receptor by zinc [65]. However, clinical studies to assess the role of zinc in treating such conditions are limited. To date, in clinical trials, the role of zinc deficiency has been assessed in various conditions such as DN, where a significant correlation between zinc deficiency, neuropathic pain severity, and CIPN was found, with a delayed pain manifestation [66,119].

Magnesium is an important mineral involved in several intracellular processes. As a metal cation, it is involved in numerous enzyme reactions and in the modulation of signal transduction via the influence of ion transport mediated by pumps, channels and carriers. This activity is principally carried out in excitable tissues [120]. Here, magnesium is able to antagonize the NMDA receptor, counteracting its activity in neuropathic pain development [121]. This led to the suggestion of magnesium supplementation as a potential therapeutic strategy against neuropathic pain. In the rat model of DN, magnesium administration caused a downregulation of NMDA receptor phosphorylation. This was followed by a reduction in allodynia [122]. Several clinical trials have been conducted but they show both negative and positive results [70,71,72,73]. In fact, neuropathic pain is not always counteracted by magnesium supplementation, either as a single treatment compound or in combination with other drugs.

### 3.3. Botanicals in Neuropathic Pain

From the Burseraceae family, *Boswellia serrata* has gained attention from researchers in recent years due to its numerous properties. *Boswellia* trees are able to release a resin called “frankincense”. This plant extract is commonly used to reduce swelling and pain caused by inflammatory disease. This is possible due to the composition of the resin. Of all the compounds contained in the *Boswellia serrata* extract, 3-O-acetyl-11-keto-beta-boswellic acid is responsible for counteracting pain [123]. The components of *Boswellia* extract are involved in several processes such as the inhibition of 5-lipoxygenase with a reduced production of leukotrienes involved in inflammatory signaling, the inhibition of NF-κB complex strongly involved in inflammation, and the downregulation or inhibition of pro-inflammatory cytokines, such as IFN-γ, TNF-α and IL-1β [124,125]. Numerous clinical studies have been conducted to assess the potential role of *Boswellia serrata* extract to counteract inflammation. The extract is a safe and fast acting compound able to exert analgesic properties in a wide spectrum of conditions. However, it is too early to establish clinical practice recommendations [76].

*Hypericum perforatum* (or St. John’s Wort) is a plant native to Europe and Asia that has been suggested as an antinociceptive for various conditions related to neuropathic pain [126]. To assess the role of this plant in neuropathic pain, several rat models have been used. In a rat model of the DN administration of *Hypericum perforatum* extract from seeds or from the aerial portion of the plant reversed mechanical hyperalgesia with an effect comparable to that of antihyperalgesic drugs [127]. Similarly, in rat model of CIPN induced with oxaliplatin, the administration of dried extract of this plant highly counteracted mechanical hyperalgesia with an effect comparable to that of antihyperalgesic drugs [127]. To date, knowledge on the *Hypericum perforatum* mechanism of action is still unclear, and several theories have been proposed. Firstly, it was shown that the antinociceptive properties are related to hypericin and hyperforin, which can be found in the extract of the plant. Another theory suggests that *Hypericum perforatum* has a mechanism in common with antihyperalgesic drugs and antidepressants [128]. In fact, the plant’s extract is primary used to treat depression [129]. In common with these two types of drugs, *Hypericum perforatum* may share the ability to inhibit monoamine oxidase and amine reuptake at synapse level [130]. From a clinical point of view, the administration of *Hypericum perforatum* extract has been involved in many studies. Despite what has been seen with preclinical data, clinical studies show ambiguous results for the efficacy of *Hypericum perforatum* in neuropathic pain patients [78,79]. This compound was also well-tolerated. It is clear that the safety of *Hypericum perforatum* is high when it is used as a monotherapy due to its adverse effects, such as drug interaction and photosensitivity [80,126]. However, solid clinical studies are needed to assess the exact role and mechanism of action of *Hypericum perforatum*.

Concerning turmeric (*Curcuma longa*), curcumin can be found in the curry spices of the ginger family. Curcumin is a lipophilic polyphenol with antioxidant and anti-inflammatory characteristics. The principal activity of curcumin is the regulation of multiple signaling pathways involved in tumor suppression, cell survival, protein kinase, caspase pathways, and others [131]. In preclinical studies, mouse models were used to assess the efficacy of curcumin in the treatment of neuropathic pain conditions. For example, in a mouse model of DN a four-week treatment with curcumin led to a reduction in the levels of TNF-α and NO, most likely related to the antinociception observed in the study. This resulted in the attenuation of thermal hyperalgesia [132]. Similarly, in a rat model of neuropathic pain, curcumin treatment caused a reversion of mechanical allodynia [133]. Moreover, in a different model, mice treated with curcumin showed a reduction in IL-1β, the JAK2-STAT3 pathway and inflammasome formation [134]. According to preclinical studies, in clinical trials, curcumin showed similar effects. In a controlled study with 160 patients under chemotherapy, curcumin was administered with an optimized delivery system (MERIVA^®^, Indena SpA, Milano, Italy) to assess its ability to prevent or counteract CIPN. This nutraceutical administration led to an overall alleviation of the CIPN condition [81]. However, the lack of clinical studies is not sufficient to counter the lack of knowledge about curcumin, and thus its role in clinics is not well-defined.

Derived from chili pepper (*Capsicum annum*), capsaicin is the compound responsible for the burning and irritant effects of chili pepper consumption. Capsaicin is related to pain. Thus, its main role is as an analgesic treatment for various disorders, including neuropathic pain conditions such as DN [135]. The mechanism in which capsaicin performs its role as an analgesic treatment relies on the ability of this phenolic compound to interact with the receptor TRPV1, directly affecting unmyelinated C-fibers that modulate calcium flux and depleting substance P, a neuropeptide involved in pain perception [136]. To date, the clinical usage of capsaicin is related to the local treatment with capsaicin-based patches, creams and lotions for daily application [137]. However, capsules containing chili peppers have been produced to allow capsaicin administration. It is important to precise that a therapeutic dosage for orally administered capsaicin is not well-defined. The available over-the-counter capsules are not for pain treatment. Therefore, a deep exploration of past literature and the creation of solid clinical trials is essential for establish capsaicin dosage in a diet supplementations therapy in order to relieve pain [138].

From a cooling molecule derived from peppermint to a common analgesic, menthol plays a critical role in pain modulation. Its major analgesic mechanism is related to activation of transient receptor potential cation channel melastatin 8 (TRPM8), involved in the hypersensitivity to cold stimuli in neuropathic pain conditions when co-expressed with TRPV1 [139]. A peculiar characteristic of TRPM8 is its role in the reduction in pain sensitivity when stimulated. However, several animal models have been used to assess the role of menthol in neuropathic pain. In those models, a reduction in pain conditions was obtained when treating animals with L-menthol isomer [140]. To date, clinical studies focused on topical administration of menthol. However, research on orally administered menthol in neuropathic pain is rare or altogether absent. An evaluation of the potential usage of menthol as an orally administered nutraceutical could be useful in order to expand the spectrum of these nutraceutical applications.

Bromelain is natural cysteine proteolytic enzyme derived from pineapple that exerts a wide range of effects such as anti-inflammatory and fibrinolytic effects, anti-cancer activity and immunomodulatory effects, wound healing, and others [141]. However, the molecular mechanism of action of bromelain have not been completely identified.

Bromelain is a safe-to-use nutraceutical that lacks side effects. Hence, its role in neuropathic pain has been assessed in preclinical and clinical studies. For example, in a sciatic-nerve ligation-induced neuropathic pain model, the treatment of rats with bromelain significantly reduced thermal hyperalgesia and mechanical allodynia, with the recovery of sciatic function and structural integrity [141]. Similarly, in another rat model of neuropathic pain, bromelain administration reduced neuropathic pain characteristic signs [141]. As for capsaicin, despite numerous preclinical studies, the lack of clinical trials is relevant. Well-defined preclinical results need to be confirmed by clinical trials in neuropathic pain conditions.

### 3.4. Amino Acids in Neuropathic Pain

Several amino acids have been suggested for the treatment of neuropathic pain conditions, such as acetyl-L-carnitine, *N*-acetyl-cysteine and glutamine. Acetyl-L-carnitine (ALC) is an endogenous molecule synthesized in the human central and peripheral nervous systems, as well as the kidneys and liver, that has very persistent analgesic effects when supplemented via diet [142]. Inside cells, this amino acid is involved in intermediary metabolism, protein acetylation, and neuroprotection. A wide range of animal models have been used to assess the neurotrophic and analgesic properties of ALC via the dietary supplementation in neuropathic pain conditions. In preclinical studies, researchers demonstrated the effect of the central anti-nociceptive action on nerve function. Thus, clinical studies tested ALC in patients with several pain conditions, in particular DN and CIPN. In double-blind randomized control trials involving 333 patients with DN, ALC treatment improved the nerve conduction velocity and amplitude [86]. On the contrary, in two randomized controlled clinical trials only involving patients with DN, no significant changes in the electrophysiology of patients were found after ALC treatment [87]. Additionally, two open-label studies with CIPN patients showed how ALC treatment may even worsen CIPN symptoms [88]. So, the results attested both positive and negative effects following ALC administration. However, the negative outcome from clinical studies is surrounded by solid preclinical studies, sustaining the high number of positive outcomes from other clinical studies. This confirms the potentiality of ALC as a dietary supplement for the treatment of neuropathic pain conditions such as DN and CIPN [143]. Additionally, a better understanding of mechanism of action and dosage focusing on the results about symptoms worsening is needed.

*N*-acetyl-cysteine (NAC) can be naturally found as diet supplement, not only in fruits and vegetables but also in grains, fish, meat, dairy and egg products [144]. NAC presents various properties such as anti-oxidant and anti-inflammatory activities related to the enhancement of Gluthathion-S-Transferase activity, the stabilization of proteins structures, mucolytic activity and others [145]. Hence, due to its properties, a potential role of NAC was suggested in the treatment of numerous diseases and conditions, including neuropathic pain. In preclinical studies of DN, NAC administration led to an amelioration of neuropathic conditions [146,147]. Preclinical results were confirmed in clinical trials. For example, in a clinical study the administration of NAC, in combination with pregabalin in DN patients, caused an overall improvement in neuropathy conditions. This effect was observed in comparison with other patients treated with pregabalin or placebo only, defining a positive role in NAC as adjuvant therapy [90]. Similarly, in patients with CIPN, another clinical study showed that the incidence and severity of this condition might be reduced by NAC oral administration [91]. Thus, the results obtained confirm the role of NAC as an analgesic and neuroprotective compound for the treatment of, for example, neuropathic pain conditions, such as DN and CIPN, in a diet-supplementation-based therapy.

### 3.5. Fatty Acids in Neuropathic Pain

ω-3 fatty acids are essential components of diet that can be obtained from fish. In the body, ω-3 are incorporated into cell membrane. Among all, eicosapentaenoic acid and docosahexaenoic acid are precursor of resolvines, inflammation mediators able to modulate pain stimulus and spinal cord plasticity. However, ω-3 are characterized by several roles such as anti-inflammatory, immunomodulatory and cardioprotective effects [148]. These properties make ω-3 a major actor in the diet supplementation approach, and this has been clinically confirmed. In a randomized double-blinded controlled trial, 57 female patients with breast cancer were evaluated. Patients were treated with paclitaxel, and the administration of ω-3 fatty acids as oral supplements caused a reduction in paclitaxel-induced neuropathic pain in terms of incidence, with the risk of 70% lowered in ω-3 treated group [1]. This defines the potential of ω-3 fatty acids to prevent CIPN. However, several clinical trials are ongoing with the aim of establishing whether ω-3 fatty acids can be useful in other neuropathic pain conditions.

Another relevant fatty acid is *N*-palmitoylethanolamide (PEA), an anandamide analogue. PEA is natural compound that can be found in egg yolks, peanut meal and soybean lecithin. It is characterized by an anti-inflammatory activity with a modulation of degranulation, reduction in TNF-α and NGF, and regulation of pain mechanisms [149,150]. In a preclinical study, PEA administration led to reduced thermal hyperalgesia and mechanical allodynia in a murine model of neuropathic pain [151]. Starting from preclinical studies, a role of vanilloid (TRPV1) and cannabinoid receptors was suggested in mediating antinociceptive mechanism of PEA. Clinically, these data were confirmed in several studies involving patients with neuropathic pain conditions. In a retrospective observational study, patients with fibromyalgia were treated with co-administered duloxetine and pregabalin, supplemented with PEA. The nutraceutical supplementation caused pain relief and a reduction in the neuropathic pain signs [94,95,152].

α-lipoic acid (ALA) is an antioxidant compound located in the mitochondria, which are involved in metabolic processes as cofactors in energy production. It can be found in many foods, such as red meats and vegetables. Due to its properties, ALA has been proposed as an antioxidant supplement. Preclinical and clinical studies confirmed ALA protective activities in models of ischemia–reperfusion lesion, neurodegeneration and neuropathy, diabetes, HIV activation, and others [153]. In particular, concerning neuropathic pain, ALA has been used to treat several related conditions, including DN and CIPN. For example, in a clinical trial, 460 patients with mild-to-moderate diabetic distal symmetric sensorimotor polyneuropathy were treated with ALA or placebo once daily for four years. ALA administration led to a clinically meaningful improvement and prevention of the progression of neuropathic pain conditions [96]. However, clinical trials in DN and CIPN are lacking. A sparse number of studies showed a non-significant improvement of neuropathic pain condition in patients orally treated with ALA [76].

### 3.6. Probiotics in Neuropathic Pain

Neuropathic pain is able to alter and cause stress in different areas of the body. To date, the lack of knowledge regarding many aspects of neuropathic pain prevents a better understanding of the many mechanisms related to its pathogenesis. In recent years, an extensive involvement of the gut in modulating pain and other aspects of neuropathic pain has been proposed [154]. Specifically, the gut–brain axis is suggested to play a relevant role in such conditions. This is due the wide spectrum of activities regulated by this axis, such as the modulation of pain, metabolic and neurological signaling, inflammation and others [155]. The major actor in the gut–brain axis is the microbiome. Previous studies showed how nociceptor neurons are capable of recognizing bacterial-secreted molecules involved in pain signaling [156]. In this context, probiotic supplements as a therapeutic strategy for the treatment of neuropathic pain conditions such as DN and CIPN have been explored. Probiotics are living bacteria able to modulate inflammatory responses affecting cytokines. By altering gut microbiota, probiotics provide health benefits such as digestion, enhanced immunity and a reduced risk of diseases [157,158]. In a model of CIPN, the probiotic formulation SLAB51 was tested in mice in order to relieve pain induced by paclitaxel and evaluate the potential application of this probiotic formulation in neurodegenerative disorders [159]. Chemotherapeutic agents strongly alter the gut microbiome, causing physiological and psychological anomalies. However, SLAB51 administration contributed to gut integrity in CIPN mice. In fact, the preservation of gut functionality and physiology led to the prevention of nerve fiber loss, a reduction in inflammation and the overall relief of neuropathic pain conditions [159]. A variety of preclinical studies demonstrated the effects of probiotics on several conditions, in particular visceral pain. In those models, *Clostridium butyricum*, *Roseburia hominis*, *Bifidobacterium infantis* 35624 and VSL#3 counteracted visceral pain in rats decreasing hypersensitivity induced by inflammation or stress [160,161]. In particular, the VSL#3 probiotic was able decrease visceral hypersensitivity by modulating the mast cell-PAR2-TRPV1 pathway [162]. From a clinical point of view, the beneficial effects of probiotics have been confirmed in chronic pain patients. The administration of *Lactobacillus reuteri* DSM 17938 in a double-blind controlled trial reduced the frequency and intensity of abdominal pain in children [97]. Therefore, the beneficial effects of probiotic administration have been well-documented during the last year. In conclusion, for the better development of personalized protocols, the identification of a patient’s microbiota profile is essential in order to develop targeted approaches for specific bacteria populations involved in neuropathic pain conditions.

## 4. Discussion

In the absence of more specific treatments for neuropathic pain, the synergistic activity between pharmacological and non-pharmacological therapies is relevant in neuropathic pain conditions, such as DN and CIPN. In this context, diet supplementation is becoming more and more important due to the affordability and the spectrum of the effects of nutraceuticals. A supplement can be defined as a product used to support something. Therefore, supporting a main therapy for a disease or a condition with a diet supplement can be a valid solution to effectively restore patients’ quality of life counteracting symptoms. Although nutritional supplements have been preclinically well studied, an overall lack of clinical trials is evident. The over-the-counter availability of nutritional supplements is rapidly increasing. Vitamins, minerals and botanicals, as well as amino acids, fatty acids and probiotic formulation, are very common compounds at everyone’s disposal. The supplements described in this review, as nutraceuticals, can be found in food, and through food their dietary intake should be at least obtained. With a balanced diet, a wide-spectrum disease prevention can be easily reached. However, when necessary, supplementation with higher dosage should be preferred. To date, researchers have principally focused on understanding, from a molecular point of view, nutraceuticals and their characteristics and properties. A great quantity of data and knowledge have been accumulated. Thus, it is necessary to conduct critical analyses of all the data to better understand whether a nutraceutical can be used as diet supplement, as a single treatment or in combination, even with other supplements: as a proposal of potential future direction. Regarding this, some groups have already started to work in this direction. Tested in patients with neuropathic pain conditions, such as CIPN, a combination of α-lipoic acid, *Boswellia serrata*, methylsulfonylmethane and bromelain in a single capsule called OPERA^®^ (GAMFARMA srl, Milan, Italy) showed an improvement of CIPN symptoms without toxicity or drug interactions [163].

## 5. Conclusions

In conclusion, it is possible to postulate that the relevant role of nutraceutical supplements has been defined by preclinical and clinical studies. The risk linked to pharmaceutical drugs administration and their side effects is leading to a change in perspective regarding the relevance of nutraceuticals supplements, enlarging their availability day by day. In this review, the role of nutritional supplements in counteracting neuropathic pain in CIPN and DN is reported, and their role as a potential treatment is raised.

## Figures and Tables

**Table 1 life-12-01125-t001:** Overview of nutritional supplements with a potential role in neuropathic pain treatment explored in this review.

Nutritional Supplements		Dietary Doses		Supplement Doses	Toxicity Level	References		
						Supplementation Effect		Other
		Male	Female			Positive	Negative	
Vitamins	B1	1.2 mg/d	1.2 mg/d	50–100 mg/d	N/A	[43,44,45,46,47,48]	[49]	[50]
	B6	1.30 mg/d	1.30 mg/d	100–130 mg/d	300–600 mg/d			[51]
	B9	400 mg/d DFE ^1^	400 mg/d DFE	400–1000 mg DFE	N/A			[52]
	B12	2.4 mg/d	2.4 mg/d	500–1000 mg	N/A			[53]
Vitamin C		90–200 mg/d	75–200 mg/d	Up to 2000 mg/d	>2000 mg/d	[54,55]	N/A	[56]
Vitamin D		600–800 IU/d	600–800 IU/d	400–5000 IU	10000 IU/d	[57,58,59,60,61,62,63]	N/A	[64]
Zinc		11 mg/d	8 mg/d	30–50 mg	100 mg/d	[65,66]	[67]	[68,69]
Magnesium		400–420 mg/d	310–320 mg/d	250–500 mg	5000 mg/d	[70,71]	[72,73]	[72]
*Boswellia* *serrata*		N/A	N/A	1200 mg/d	1000 mg/kg	[74,75]	N/A	[76,77]
*Hypericum perforatum*		N/A	N/A	100–900 mg	Drug interactions	[78]	[79,80]	[79]
Curcumin		N/A	N/A	500–1000 mg	8000 g/d	[81]	N/A	[82]
Capsaicin		0.25%	0.25%	N/A	>8%	N/A	N/A	[83]
Menthol		N/A	N/A	180–400 mg 3 times daily	1000 mg/kg	N/A	N/A	[84]
Bromelain		N/A	N/A	80–300 mg 2 or 3 times daily	N/A	N/A	N/A	[85]
Acetyl-L-Carnitine		250–330 mg	250–330 mg	1500–3000 mg/d	>3000 mg/d	[86]	[87,88]	[89]
*N*-Acetyl-Cysteine		400–1200 mg/d	400–1200 mg/d	<5000 mg/d	>5000 mg/d	[90,91]	N/A	[92]
ω-3 fatty acids		1600 mg/d	1100 mg/d	2000–4000 mg/d	>4000 mg/d	[1]	N/A	[93]
*N*-palmitoylethanolamide		1200–1600 mg/d	1200–1600 mg/d	300–2400 mg/d	N/A	[94,95]	N/A	[76]
α-lipoic acid		200–1800 mg/d	200–1800 mg/d	1000–2400 mg/d	N/A	[96]	N/A	[76]
Probiotics		N/A	N/A	1 to 10 billion CFU	>50 billion CFU	[97]	N/A	[98,99]

^1^ DFE, Dietary folate units.

## Data Availability

Not applicable.

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
