# Peer review of "The Influence of Dietary Supplementations on Neuropathic Pain"

_life, 2022, doi:10.3390/life12081125_

Round 1

Reviewer 1 Report

Due to the lack of clinical trials, the role and effects of many supplement or nutraceuticals compounds on neuropathic pain is difficult to be defined .

Detailed comments are as follows:

1. This study reach his aim on the synergistic activity between pharmacological and non pharmacological therapies , so relevant in dn and cipn - could you put arachidonic acid leading to cascade of pro infiammatory cytokines th1 th6 th8 e tnf?

2. the topic is  original and relevant in the field in order to restore patients' quality of life.

3. Methodology is good but there is lack of trials - could you stress?

Author Response

Due to the lack of clinical trials, the role and effects of many supplement or nutraceuticals compounds on neuropathic pain is difficult to be defined .

Response: We would like to thank Reviewer 1 for the positive comments and the time spent in reading our manuscript and for the comments provided that helped in improving our article. We tried to address all the points raised.

Detailed comments are as follows:

  1. This study reach his aim on the synergistic activity between pharmacological and non pharmacological therapies , so relevant in dn and cipn - could you put arachidonic acid leading to cascade of pro infiammatory cytokines th1 th6 th8 e tnf?

Response: We appreciate the reviewer’s comments and added this information as suggested. Please see lines 161-170.

  1. the topic is original and relevant in the field in order to restore patients' quality of life.
  2. Methodology is good but there is lack of trials - could you stress?

Response: We appreciate the reviewer’s comments, and we totally agree. We now added a few clinical trials performed so far and we stressed this part as suggested. We now highlighted the lack of clinical trials in this topic.

Reviewer 2 Report

# life-1825099

 “The influence of diet on neuropathic pain”.

This review is basically well written review manuscript concerning interaction between nutritional supplements or nutraceuticals on the neuropathic pain including preclinical and clinical evidence. Although author should appropriately address the some issues, this reviewer recommend that the manuscript is worth of publication for Life. Authors should be addressed some issues.

 Comments

1.     Introduction 1st paragraph of later part, “Among them, the therapy… counteracting neuropathic pain: Authors should describe evidence for interaction between nutritional supplements or nutraceuticals on the neuropathic pain, including appropriate paper citation. 

2.     Discussion paragraph, Authors should describe in more detail for nutritional supplements in neuropathic pain and also comment on future direction for reader.

 3.     Conclusion, Since the conclusion section is unclear, authors should  revise.

Author Response

This review is basically well written review manuscript concerning interaction between nutritional supplements or nutraceuticals on the neuropathic pain including preclinical and clinical evidence. Although author should appropriately address the some issues, this reviewer recommend that the manuscript is worth of publication for Life. Authors should be addressed some issues.

Response: We would like to thank Reviewer 2 for the positive comments and the time spent in reading our manuscript and for the comments provided that helped in improving our article. We tried to address all the points raised.

Comments

  1. Introduction 1stparagraph of later part, “Among them, the therapy… counteracting neuropathic pain: Authors should describe evidence for interaction between nutritional supplements or nutraceuticals on the neuropathic pain, including appropriate paper citation. 

Response: We appreciate the Reviewer’s comment and we now provided evidence for interaction between nutritional supplements or nutraceuticals on the neuropathic pain as suggested. Please see lines 46-50.

  1. Discussion paragraph, Authors should describe in more detail for nutritional supplements in neuropathic pain and also comment on future direction for reader.

Response: We totally agree with the Reviewer, and we now added more details as suggested. Please see lines 551-557.

  1. Conclusion, Since the conclusion section is unclear, authors should  revise.

Response: Thank you for the comment. We now revised also this part, please see 565-569 lines.

Reviewer 3 Report

The present manuscript is of potential interest due to the important subject related to chronic pain treatment, however it shows some problems that need to be addressed. In first place in the pathophysiology session there are some data not correct regarding the pain pathways, all the session need to be carefully revised!

Title: in the title the use of the word DIET is not correct since it refers generally to food not to nutraceuticals separated from the original food, indeed this word is rarelly used in the manuscript.

Due to the contrasting result present for some compounds it would be better to include a table with both positive and negative findings.

A minor point is the presence of abbreviations that need to be avoided when necessary

Author Response

The present manuscript is of potential interest due to the important subject related to chronic pain treatment, however it shows some problems that need to be addressed. In first place in the pathophysiology session there are some data not correct regarding the pain pathways, all the session need to be carefully revised!

Title: in the title the use of the word DIET is not correct since it refers generally to food not to nutraceuticals separated from the original food, indeed this word is rarelly used in the manuscript.

Response: We would like to thank Reviewer 3 for the positive comments and the time spent in reading our manuscript and for the comments provided that helped in improving our article. We agree with the Reviewer and now changed the title as suggested. We tried to address all the points raised.

Due to the contrasting result present for some compounds it would be better to include a table with both positive and negative findings.

Response: We appreciate the Reviewer’s comment and we agree. We now modified the table accordingly.

A minor point is the presence of abbreviations that need to be avoided when necessary

Response: Thank you for the comment, we tried to reduce unnecessary abbreviations.

Round 2

Reviewer 3 Report

The presynaptic inhibition described in the text about the descending modulation of nociceptive afferents is not mediated by GABA but from enkephalin. This need to be corrected

Moreover, the glutamate- induced activation of the NMDA channels is NOT automatic since it needs EPSP to be generated able to move Mg from the channel

I think that also basic information about the pain system need to be given correctly

Author Response

Reviewer 3 (Second Round)

The presynaptic inhibition described in the text about the descending modulation of nociceptive afferents is not mediated by GABA but from enkephalin. This need to be corrected

Response: We apologize for the oversight and modified this section accordingly. Please see lines: 140-147.

Moreover, the glutamate- induced activation of the NMDA channels is NOT automatic since it needs EPSP to be generated able to move Mg from the channel. I think that also basic information about the pain system need to be given correctly.

Response: We appreciate the Reviewer’s comment and we agree. We now revised the text. Please see lines: 130-134.

We would like to thank the reviewer for the valuable comments, and we hope that now the manuscript is suitable for publication.
